# An Empirical Model to Estimate Abundance of Nanophase Metallic Iron (npFe$^0$) in Lunar Soils

**Dawei Liu [1,\*], Yuanzhi Zhang [1,2], Guangliang Zhang [1], Bin Liu [1], Xin Ren [1], Rui Xu [3] and Chunlai Li [1]**

[1]   Key Laboratory of Lunar and Deep Space Exploration, National Astronomical Observatories, Chinese Academy of Sciences, Beijing 100101, China; zhangyz@nao.cas.cn (Y.Z.); zhanggl@nao.cas.cn (G.Z.); liub@nao.cas.cn (B.L.); renx@nao.cas.cn (X.R.); licl@nao.cas.cn (C.L.)

[2]   School of Astronomy and Space Science, University of Chinese Academy of Sciences, Beijing 100049, China

[3]   Key Laboratory of Space Active Opto-Electronics Technology, Shanghai Institute of Technical Physics, Chinese Academy of Sciences, Shanghai 200083, China; xurui@sitp.ac.cn

\*   Correspondence: liudw@nao.cas.cn

**Abstract:** Lunar soils gradually become mature when they are exposed to a space environment, and nanophase metallic iron (npFe$^0$) generates within them. npFe$^0$ significantly changes the optical properties of lunar soils and affects the interpretation of the remotely sensed data of the lunar surface. In this study, a correlation analysis was conducted between npFe$^0$ abundance and reflectance spectra at short wavelengths for lunar soil samples in four size groups based on their spectral and compositional data, collected by the Lunar Soil Characterization Consortium (LSCC). Results show that 540 nm single scattering albedo (SSA) of lunar soils correlates well with their corresponding npFe$^0$ abundance for each size group of lunar soil samples. However, it is poorly correlated with npFe$^0$ abundance when all size groups were considered because of the strong interference from grain size variation of lunar soils. To minimize the effect of grain size, the correlation of npFe$^0$ abundance with the spectral ratio of 540 nm/810 nm SSA of all size groups for LSCC samples was calculated and results show that a higher correlation existed between them ($R^2 = 0.91$). This ratio can serve as a simple empirical model for estimating npFe$^0$ abundance in lunar soils. However, bias could be introduced to the estimation result when lunar soils possess a high content of agglutinitic glass and ilmenite. Our future work will focus on improving the model's performance for these lunar soils.

**Keywords:** npFe$^0$; single scattering albedo; lunar soil characterization consortium

## 1. Introduction

Space weathering represents any natural processes pursuing an airless planetary body that largely modify its surface material's physical and compositional characteristics [1,2]. It is mainly induced by solar wind ion implantation and micrometeoroid bombardment and has been generally applied to planetary observations, such as on the Moon or Mercury as well as asteroids [3–6]. Space weathering also influences the reflectance spectra of planetary surface materials and reduces our capability to remotely evaluate the planetary mineralogy and composition on their surface [1,7].

Samples from the Moon and asteroids provide us an opportunity to research space weathering of an airless body. Actually, when the Apollo soil samples were brought back to the Earth-based laboratories, the measured results of lunar soils quickly showed that their reflectance is very different from that of pulverized lunar rocks [7–9]. In comparison with the optical properties of pulverized lunar rocks, the spectral reflectance of lunar soils shows a red-sloped continuum (namely reddening), a lower albedo (namely darkening), and attenuated absorption bands [1,10]. Early literature attributed

these optical changes to the presence of dark amorphous "glass" or "agglutinate" in lunar soils [11]. However, subsequent laboratory and remote sensing measurement have showed that these optical properties of lunar soils are caused by the accumulation of fine grained nanophase metallic iron ($npFe^0$) residing in amorphous rims of lunar soil grains after a long duration exposure of fresh lunar soils to a space weathering environment [12–16]. Two major mechanisms have been proposed to explain the accumulation of $npFe^0$ within lunar soils. One is solar wind ion implantation stating that surface material of lunar soil grains such as O atoms are preferentially sputtered from their lattice sites as a result of ion-irradiation, generating a reducing environment in which iron particles within residue surface material are reduced to form $npFe^0$ [10,17–19]. The other mechanism proposed that because of continuous impact of micrometeoroids, lunar soil grains are vaporized so that some FeO molecules are dissociated into their constituent neutral atoms [10,20]. The O atoms dissipate, and the iron atoms deposit and accumulate on the surface of lunar soil grains as $npFe^0$ [21]. A number of laboratory experiments have been carried out to simulate the process of solar wind bombardment and micrometeorite impact events to investigate space weathering and the formation of $npFe^0$ [21–28]. For example, Duke et al. [22] irradiated an olivine sample with 1.0 keV $H^+$ and 4 keV $He^+$ to simulate solar wind implantation and found that the surface of iron of olivine is dramatically reduced to the metallic form. The process of micrometeorite impacts was simulated by [25,26] using pulse-laser irradiation on the olivine and pyroxene grains. Results show that lunar-like $npFe^0$ could form within the amorphous rims of mineral grains and the darkening and reddening effect on the spectra of these grains can be observed after pulse-laser irradiation. By performing slow- and rapid-heating experiments on mature lunar soils, Thompson et al. [21] simulated micrometeorite impact events and provided the first direct in-situ observation of $npFe^0$ formation within lunar soil grains. In addition to the generation of $npFe^0$, recent work also suggests that nanoparticles of $Fe^{2+}$ and $Fe^{3+}$ oxidation states could also form within sub-mature and mature lunar soils as a result of space weathering [29].

Estimation of $npFe^0$ abundance in lunar soils is a very interesting topic for many lunar scientists. It is helpful to accurately evaluate the lunar surface's mineralogy and composition using remotely sensed measurement. Currently, Hapke's radiative transfer model (RTM) has been generally applied to assess the mineralogy of the lunar surface owing to its ability to account for the effects of space weathering. Estimation of $npFe^0$ abundance in lunar soils is a necessary critical input for accurate Hapke's RTM mineral abundance inversion. The $npFe^0$ has been also associated with the presence of $H_2O/OH^-$ due to reduction of $Fe^{2+}$ to $npFe^0$ by impinging solar wind with the release of $H_2O/OH^-$ in lunar soils [30]. This brings about a highly possible usage of $npFe^0$ to determine lunar surface regions with $H_2O/OH^-$ [31]. In addition, determination of the abundance of $npFe^0$ in lunar soils helps us to understand the formation of lunar swirls, which were once attributed to deficient $npFe^0$ as a result of retarded space weathering [32–34].

Despite the significant importance of $npFe^0$, its abundance in lunar soils has not been widely quantitatively evaluated [35–38]. Some previous studies attempted to establish models that can relate the measured reflectance spectra of lunar soils to their characteristic ferromagnetic resonance ($I_s$) induced by $npFe^0$. $I_s$ is highly linearly correlated with the abundance of $npFe^0$ and can be used as a measure of $npFe^0$ in lunar soils [39–41]. According to the measured reflectance spectra and $I_s$ of 35 lunar soil samples, Hiroi et al. [35] found that $I_s$ shows a good linear correlation with the spectral slope defined by a straight-line tangent to the shoulders of a 1 µm absorption band of lunar soils. Mouélic et al. [36] expanded the number of investigated lunar samples to 50 and found that $I_s$ can be also evaluated by the continuum slope derived from the ratio of 1500 nm/750 nm reflectance. Pieters et al. [37] developed several statistically optimized formulations that can link $I_s$ of lunar soils with their corresponding Clementine-bands-based spectral parameter according to the compositional and spectral data acquired by the Lunar Soil Characterization Consortium (LSCC). Compared with the work of [35,36], more samples of lunar soil were used in their investigation and the sample dataset shows a larger variation in composition and maturity which is well suited for characterizing the spectral properties of lunar soils. Recent work estimating the abundance of $npFe^0$ in lunar soils has

been done by Trang and Lucey [38], who derived the npFe$^0$ abundance by using Hapke's RTM and constraining the input FeO and ilmenite content of lunar soils.

While these previous studies have provided possible ways to predict npFe$^0$ or I$_s$ of lunar soils, the practical use of these methods will be impeded by several limitations. For example, most of these works require band information at longer wavelength (e.g., 1500 nm) to define the spectral slope [35,36], which is difficult in some cases. On the one hand, the spectral range of many remote sensing sensors stops at around 1 μm and no longer-wavelength data are available (e.g., Clementine, Chang'E-1 Interference Imaging Spectrometer). On the other hand, it is a challenge to accurately define a continuum slope when lunar soils become highly mature or lack apparent absorption features at around 1 μm [35]. In addition, the model for npFe$^0$ estimation could be too complex to be applied. Many input parameters and pre-knowledge on the compositional information of lunar soils are required before the model can be used, which is difficult to obtain in advance [38].

Here, we developed a simple empirical model to estimate npFe$^0$ abundance in lunar soils from measured reflectance data. This model correlates the npFe$^0$ abundance with the ratio of single scattering albedo (SSA) of two wavelengths outside the absorption bands of lunar soils and requires no longer-wavelength spectral data. Grain size effects of lunar soils on npFe$^0$ abundance estimation are partially removed according to this model.

## 2. Dataset

The dataset of the Lunar Soil Characterization Consortium (LSCC) [7,15] was applied in the study to develop the model. The dataset includes 10 highland and 9 mare soil samples. Each soil sample was grouped into four types of grain sizes (< 10 μm, 10–20 μm, 20–45 μm, and < 45 μm). The maturity index (I$_s$/FeO), composition (e.g. FeO content) and reflectance spectra of each size group (altogether 76 data samples) were measured by the LSCC (Table 1). The spectral data of these soil samples can be downloaded from Reflectance Experiment Laboratory (Relab) of Brown University (http://www.planetary.brown.edu/relabdocs/LSCCsoil.html). The maturity index, composition and mineralogy information of corresponding soil samples can be acquired from the Planetary Geoscience Institute of the University of Tennessee (https://pgi.utk.edu/lunar-soil-characterization-consortium-lscc-data/). Correlation was analyzed for all the samples except for sample 71061 in that it is abundant in black beads [42], with very different optical properties from those of other samples [15,43].

**Table 1.** Lunar Soil Characterization Consortium (LSCC) dataset used in this study.

| Sample | FeO (%) | I$_s$/FeO | Ilmenite (%) | Agglutinitic Glass (%) | Grain Size | Highland/Mare |
|---|---|---|---|---|---|---|
| 10084 | 12 | 145 | 5 | 62.6 | <10 μm | Mare |
| 12001 | 12.5 | 115 | 1.6 | 61.9 | <10 μm | Mare |
| 12030 | 14.3 | 32 | 3 | 55 | <10 μm | Mare |
| 15041 | 11 | 161 | 0.7 | 70.4 | <10 μm | Mare |
| 15071 | 9.59 | 159 | 1.2 | 59.7 | <10 μm | Mare |
| 70181 | 12.7 | 104 | 3.4 | 58.3 | <10 μm | Mare |
| 71501 | 13.5 | 88 | 7.6 | 53.1 | <10 μm | Mare |
| 79221 | 11.3 | 169 | 5.2 | 61.5 | <10 μm | Mare |
| 14141 | 7.66 | 14.5 | 1.7 | 45.9 | <10 μm | Highland |
| 14163 | 8.83 | 87 | 1.1 | 66.3 | <10 μm | Highland |
| 14259 | 7.82 | 174.8 | 1.5 | 71.6 | <10 μm | Highland |
| 14260 | 8.1 | 144.9 | 1.3 | 66.5 | <10 μm | Highland |
| 61141 | 3.66 | 119.3 | 0.3 | 61.6 | <10 μm | Highland |
| 61221 | 3.64 | 19.8 | 0.9 | 41.6 | <10 μm | Highland |
| 62231 | 3.63 | 169 | 0.4 | 69.5 | <10 μm | Highland |
| 64801 | 3.84 | 115.2 | 0.2 | 63.6 | <10 μm | Highland |
| 67461 | 3.35 | 35.2 | 0.2 | 35.8 | <10 μm | Highland |

**Table 1.** *Cont.*

| Sample | FeO (%) | $I_s$/FeO | Ilmenite (%) | Agglutinitic Glass (%) | Grain Size | Highland/Mare |
|---|---|---|---|---|---|---|
| 67481 | 3.61 | 38.5 | 0.2 | 35.2 | <10 μm | Highland |
| 10084 | 14.7 | 87 | 5.2 | 57 | 10–20 μm | Mare |
| 12001 | 15.9 | 67 | 1.8 | 56.8 | 10–20 μm | Mare |
| 12030 | 17.2 | 17 | 3.2 | 49.8 | 10–20 μm | Mare |
| 15041 | 14.4 | 92 | 0.8 | 56.7 | 10–20 μm | Mare |
| 15071 | 15.4 | 80 | 1.8 | 49.2 | 10–20 μm | Mare |
| 70181 | 15.5 | 63 | 6.7 | 51.7 | 10–20 μm | Mare |
| 71501 | 16.4 | 50 | 9.7 | 44.8 | 10–20 μm | Mare |
| 79221 | 15 | 78 | 6 | 54.3 | 10–20 μm | Mare |
| 14141 | 9.46 | 11.6 | 1.1 | 48.6 | 10–20 μm | Highland |
| 14163 | 10.1 | 64.8 | 0.9 | 58.5 | 10–20 μm | Highland |
| 14259 | 9.71 | 101.8 | 1.2 | 68.7 | 10–20 μm | Highland |
| 14260 | 9.84 | 98.9 | 1 | 65.2 | 10–20 μm | Highland |
| 61141 | 5.14 | 81.6 | 0.3 | 53.9 | 10–20 μm | Highland |
| 61221 | 4.4 | 13.89 | 0.3 | 32.6 | 10–20 μm | Highland |
| 62231 | 4.86 | 109.9 | 0.5 | 55 | 10–20 μm | Highland |
| 64801 | 4.78 | 84.9 | 0.2 | 61 | 10–20 μm | Highland |
| 67461 | 4.64 | 23.9 | 0.3 | 32.4 | 10–20 μm | Highland |
| 67481 | 4.04 | 33 | 0.2 | 28.6 | 10–20 μm | Highland |
| 10084 | 15.5 | 67 | 6.4 | 53.9 | 20–45 μm | Mare |
| 12001 | 16.9 | 51 | 2.6 | 56.2 | 20–45 μm | Mare |
| 12030 | 17.6 | 12 | 2.6 | 39.4 | 20–45 μm | Mare |
| 15041 | 15.2 | 66 | 1.2 | 51.3 | 20–45 μm | Mare |
| 15071 | 15.6 | 49 | 1.9 | 47.6 | 20–45 μm | Mare |
| 70181 | 16 | 53 | 8.9 | 43.4 | 20–45 μm | Mare |
| 71501 | 17.8 | 28 | 12.3 | 38.3 | 20–45 μm | Mare |
| 79221 | 15.8 | 57 | 7.3 | 46.5 | 20–45 μm | Mare |
| 14141 | 11.6 | 5.8 | 1.9 | 41 | 20–45 μm | Highland |
| 14163 | 11.5 | 43.2 | 0.8 | 56.4 | 20–45 μm | Highland |
| 14259 | 11 | 77.2 | 1.3 | 60.5 | 20–45 μm | Highland |
| 14260 | 10.7 | 80.2 | 0.9 | 64 | 20-45 μm | Highland |
| 61141 | 5.15 | 75.5 | 0.3 | 50.1 | 20–45 μm | Highland |
| 61221 | 4.62 | 8.4 | 0.6 | 28.9 | 20–45 μm | Highland |
| 62231 | 5.31 | 80.7 | 0.3 | 50.6 | 20–45 μm | Highland |
| 64801 | 4.82 | 83.4 | 0.3 | 53.6 | 20–45 μm | Highland |
| 67461 | 4.93 | 22.3 | 0.3 | 25.4 | 20–45 μm | Highland |
| 67481 | 5.19 | 20.7 | 0.1 | 27.6 | 20–45 μm | Highland |
| 10084 | 14.8 | 88 | | | <45 μm | Mare |
| 12001 | 16 | 62 | | | <45 μm | Mare |
| 12030 | 16 | 20 | | | <45 μm | Mare |
| 15041 | 14.2 | 93 | | | <45 μm | Mare |
| 15071 | 14.9 | 71 | | | <45 μm | Mare |
| 70181 | 15.3 | 61 | | | <45 μm | Mare |
| 71501 | 16.5 | 44 | | | <45 μm | Mare |
| 79221 | 14 | 91 | | | <45 μm | Mare |
| 14141 | 9.81 | 9.7 | | | <45 μm | Highland |
| 14163 | 9.94 | 66.5 | | | <45 μm | Highland |
| 14259 | 9.54 | 108.6 | | | <45 μm | Highland |
| 14260 | 9.65 | 93.3 | | | <45 μm | Highland |
| 61141 | 4.8 | 94.5 | | | <45 μm | Highland |
| 61221 | 4.47 | 13.6 | | | <45 μm | Highland |
| 62231 | 4.87 | 116.7 | | | <45 μm | Highland |
| 64801 | 4.68 | 92.2 | | | <45 μm | Highland |
| 67461 | 4.24 | 29.8 | | | <45 μm | Highland |
| 67481 | 4.38 | 33.5 | | | <45 μm | Highland |

Note: agglutinitic glass and ilmenite contents for size group < 45 μm are not provided in LSCC dataset

## 3. Method

### 3.1. Strong Influence of npFe$^0$ at Short Wavelengths

Liu et al. and Liu. [44,45] conducted sensitivity analyses of a suite of Hapke's RTM-simulated reflectance spectra of lunar soils with the aim of quantitatively determining the relative importance of npFe$^0$ abundance, grain size and mineral abundance in controlling the optical properties of lunar soils. Results show that npFe$^0$ has strong influences in the full spectral ranges as compared to mineral abundance. The simulated spectral reflectance of lunar soils is more sensitive to the variation of npFe$^0$ at shorter spectral regions than longer spectral ranges. The strong npFe$^0$ effect on short-wave bands is evident in the spectral reflectance of LSCC data, which tend to be convergent in short spectral regions for different sizes of grains in each sample of lunar soils (Figure 1). This indicates that the spectral reflectance of lunar soils in short wavelengths is less sensitive to the variation of composing minerals of lunar soils, while mainly controlled by the variation of npFe$^0$ abundance [44,45]. The strong influence of npFe$^0$ at short wavelengths can be also demonstrated by the work of [46], who prepared a suite of analog soils of varying abundance of npFe$^0$ to study the optical effects of npFe$^0$ on the measured reflectance spectra. Result shows that npFe$^0$ (<10 nm) greatly reddens short wavelengths spectra while leaving the longer wavelengths spectra mostly unaffected. Using lunar samples of similar composition but distinct exposure age, Fischer et al. and Fischer and Pieters [47,48] attempted to develop a mathematical model that can describe the optical alteration of lunar soils due to space weathering. Their work suggests that the effect of npFe$^0$ on the optical properties of lunar soils can be evaluated as a function of wavelength with an increasing rate of darkening toward short wavelengths. This result is also consistent with Mie theory, which states that if the npFe$^0$ is smaller than the wavelength of incident light, the greatest amount of light extinction will be at the short wavelengths region [48]. Combing Hapke's RTM and Mie theory to model the scattering behavior of npFe$^0$, Wohlfarth et al. [49] also found a strong influence of npFe$^0$ at short wavelengths of simulated reflectance spectra.

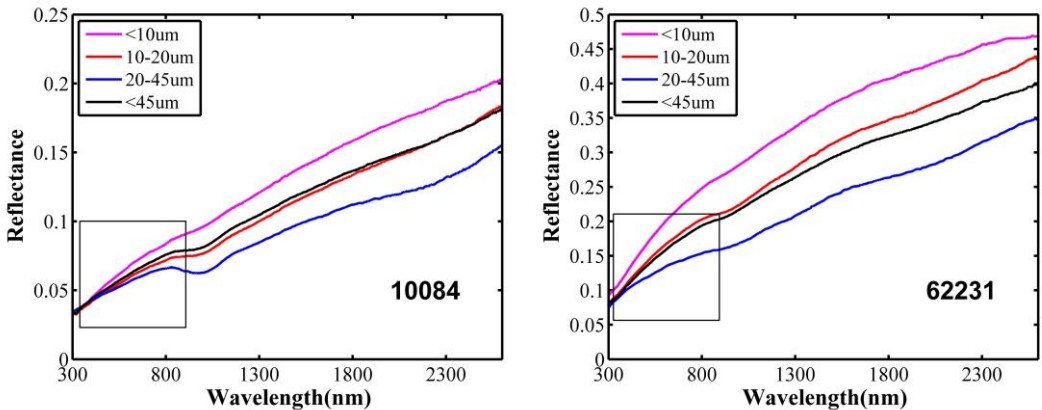

**Figure 1.** Reflectance spectra of four grain size groups of lunar soil samples 10084 and 62231. Black squares show the convergence of reflectance spectra at short wavelengths.

### 3.2. Estimation of npFe$^0$ Abundance in Lunar Soils

The strong effect of npFe$^0$ on short wavelengths spectra motivates us to explore the correlation between the npFe$^0$ abundance of lunar soils and their spectral of corresponding reflectance values only at short spectral ranges. Through conducting ferromagnetic resonance measurement on lunar soil samples, Morris [41] found a linear relation among npFe$^0$ abundance, FeO content and maturity index ($I_s$/FeO), reproduced below:

$$npFe^0 = \left(3.2 \times 10^{-4}\right)(FeO)(I_s/FeO) \tag{1}$$

In this study, the abundance of npFe$^0$ for each lunar sample was first obtained via Equation (1) based on the measured maturity index ($I_s$/FeO) and FeO content (Table 1). Then, the LSCC measured

spectral reflectance of each sample was re-sampled to the same bands as Moon Mineralogy Mapper ($M^3$) data and transformed to single scattering albedo (SSA) using a simplified Hapke's RTM (Appendix A and Equation (A2) of [50]). This transformation has potential to remove the effects of multiple scattering because npFe$^0$ residing in the rims of soil grains mainly influences SSA of lunar soils. Considering the influence of npFe$^0$ increases with decreasing wavelength [44,45], 540 nm SSA, which is the first available band of $M^3$ data of high quality, was chosen for the correlation analysis with the obtained npFe$^0$ abundance. Finally, the correlation of npFe$^0$ abundance with the spectral ratio of 540 nm SSA/810 nm SSA was also calculated, aiming at minimizing the effect of grain size of lunar soils on npFe$^0$ estimation. The band at 810 nm was selected because it is outside the absorption wavelength regions of lunar soils, and the influence of composing minerals on npFe$^0$ estimation could be minimized.

## 4. Result

### 4.1. Correlation between 540 nm Single Scattering albedo (SSA) and npFe$^0$

Figure 2 shows the correlation of npFe$^0$ abundance with 540 nm SSA of all samples of LSCC soils. In general, as the abundance of npFe$^0$ increases, the SSA of the corresponding lunar soil decreases, which is consistent with the darkening effect of npFe$^0$, as expected. However, for all groups of investigated soil samples, the npFe$^0$ does not show a good correlation with the corresponding SSA at 540 nm. This could be attributed to the influence of grain size variation of lunar soils at short wavelengths. In addition to the varying abundance of npFe$^0$, the grain size of lunar soils is also a significant factor driving the measured reflectance. For example, the spectral reflectance of a suite of transparent materials with different sizes of grains was measured by Adams and Filice [51] and Pieters [52], whereby it was noted that there is a general increase in the overall reflected spectra with decreasing grain size.

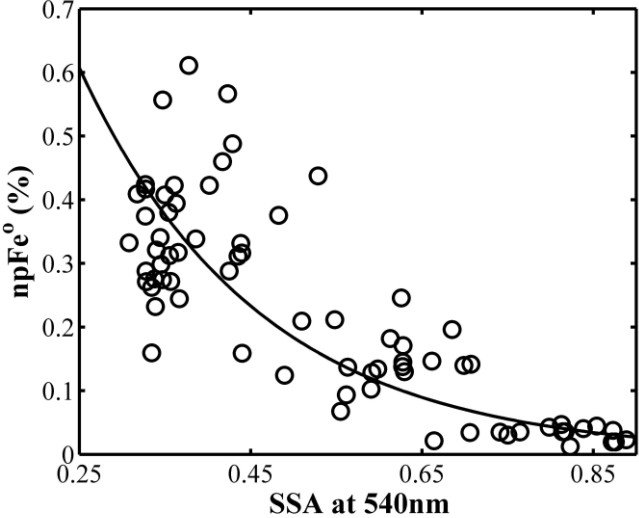

**Figure 2.** Correlation of SSA at 540 nm with npFe$^0$ for all size groups of LSCC soils.

To validate this speculation, correlation was analyzed for each group of sizes of LSCC samples seen in Figure 3. It is clear that the npFe$^0$ of each size group highly correlates to the SSA at 540 nm, with the determination ($R^2$) close to or higher than 0.90 (Table 2). The abundance of npFe$^0$ in each size group of LSCC data can be well fitted in the form of an exponential function as $Y = \alpha e^{(-\beta X)}$. Here, X is SSA at 540 nm, Y is the abundance of npFe$^0$, and $\alpha$ and $\beta$ are two regressed constants. Detailed values of $\alpha$, $\beta$ and $R^2$ for each size group samples are referred to Table 2. The regression lines for 10–20 μm and the < 45 μm size groups have similar $\alpha$ and $\beta$ (Table 2) because the reflectance spectra of bulk soil (< 45 μm) are most similar to that of a 10–20 μm size fraction [1] (Figure 1).

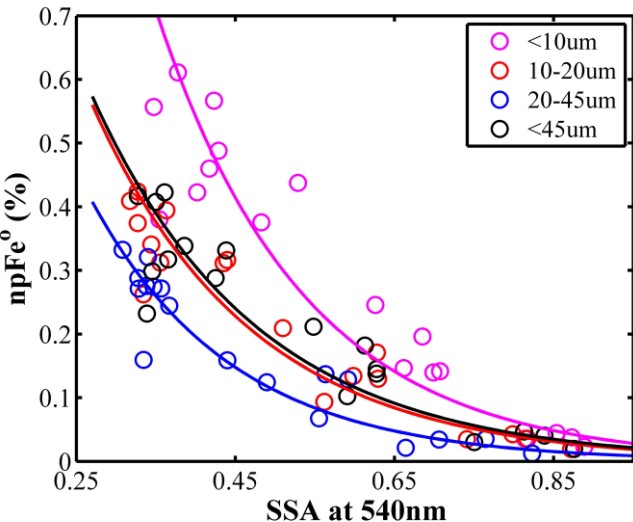

**Figure 3.** Correlation of SSA at 540 nm and npFe$^0$ for each grain size group of LSCC samples. Solid color lines represent regression lines for different size groups of lunar samples

**Table 2.** Model parameters and R$^2$ for each grain size group of LSCC data.

| Grain Size | $\alpha$ | $\beta$ | R$^2$ |
| --- | --- | --- | --- |
| <10 μm | 4.6478 | 5.375 | 0.90 |
| 10–20 μm | 2.1549 | 4.991 | 0.92 |
| 20–45 μm | 1.8737 | 5.648 | 0.89 |
| <45 μm | 2.1155 | 4.836 | 0.89 |

### *4.2. Correlation between the Ratio of 540 nm SSA/810 nm SSA and npFe$^0$*

To minimize the effect of grain size, the correlation of the spectral ratio of 540 nm/810 nm SSA of all size groups of LSCC samples was calculated with npFe$^0$. It clearly reveals that a higher correlation was obtained with R$^2$ up to 0.91 (Figure 4) after normalizing 540 nm SSA by 810 nm SSA. Much less scattering is shown in Figure 4 than in Figure 3, which demonstrates the success of using the spectral ratio of 540 nm/810 nm SSA in minimizing the grain size effect. This pronounced improvement in npFe$^0$ estimation can be accounted for by several reasons. First, grain size has a significant influence on reflectance spectra at all wavelength regions [44,45,52]. By normalizing SSA at 540 nm to that at 810 nm, the effects of grain size could be significantly cancelled out. Second, both 540 nm and 810 nm are outside the absorption wavelength regions of lunar soils and cannot be largely affected by the Fe$^{2+}$ absorption of lunar mafic minerals. Third, the ratio of 540 nm SSA over 810 nm SSA can be regarded as a slope measure for the reflected continuum at short spectral regions; the higher the npFe$^0$ included in a lunar soil is the redder the slope in the reflected continuum is, and vice versa.

Our result also suggests that this simple SSA ratio is suited for npFe$^0$ estimation of lunar soils showing a wide range of maturity (fresh, sub-mature, mature). Even for the mature lunar samples (I$_s$/FeO > 60), variation of npFe$^0$ can to some degree be reflected by the variation of the 540 nm/810 nm SSA ratio (Figure 5).

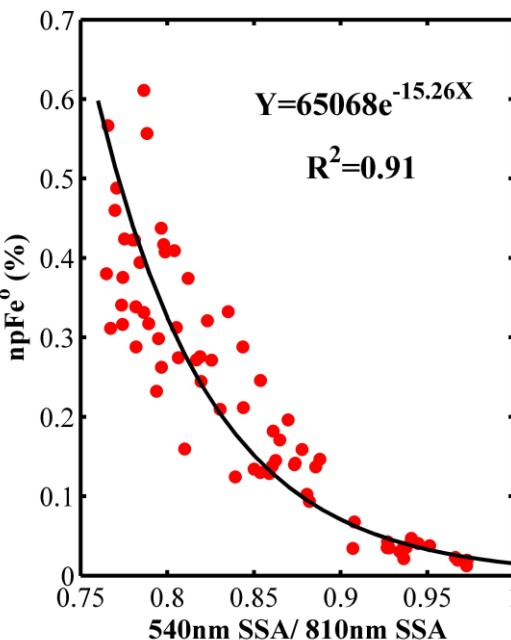

**Figure 4.** Correlation between the ratio of 540 nm SSA/810 nm SSA and npFe$^0$ for all size groups of LSCC soil samples.

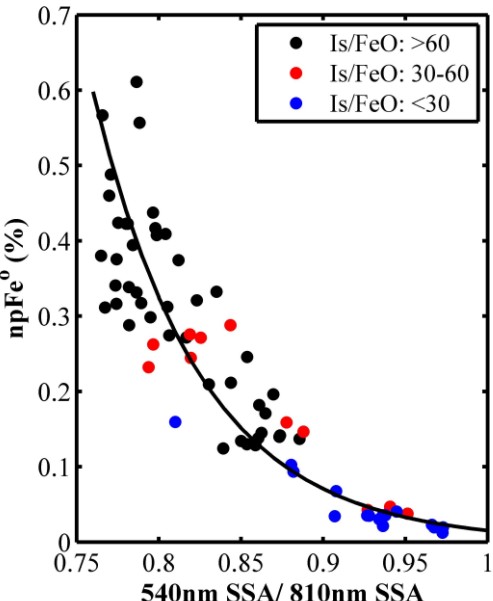

**Figure 5.** Correlation between the ratio of 540 nm SSA/810 nm SSA and npFe$^0$. Different solid color circles represent different maturity groups ($I_s$/FeO) of LSCC lunar samples. The black line is the regression line defined in Figure 4.

Although this empirical relationship was developed on the basis of LSCC data samples with a grain size < 45 μm, it might be also suited for estimating the npFe$^0$ of lunar soils with a larger grain size. Pieters et al. [1] measured the spectra of lunar soils as a function of grain size and found that while these lunar soils are extremely poorly sorted (varying from several microns to several hundred microns), the spectra of the bulk of the lunar soils were most similar to that of the finest size fraction (< 45 μm). The optical properties of lunar soils are dominated by this narrow size range because of their disproportionately large surface areas [1,53].

We also investigated the correlation between the $npFe^0$ and the ratio of 540 nm to 810 nm reflectance rather than SSA for all groups of LSCC samples (Figure 6). It can be seen that abundance of $npFe^0$ is poorly correlated with the ratio of 540 nm/810 nm reflectance of lunar soils. This is not unexpected because SSA is used to describe the scattering behavior of a single particle of lunar soil. $npFe^0$ in the amorphous rims of soil grains significantly influences a single particle's capability in reflecting or absorbing the incident light [10] and causes the SSA of lunar soils to be highly sensitive to the variation of $npFe^0$. However, in addition to SSA, the measured reflectance spectra of lunar soils are also influenced by other factors, such as multiple scattering and opposition effect [54], resulting in the unsensitivity of lunar soil spectra to the variation of $npFe^0$. Therefore, it is necessary to covert remotely obtained reflectance spectra to their corresponding SSA to estimate $npFe^0$ abundance in lunar soils.

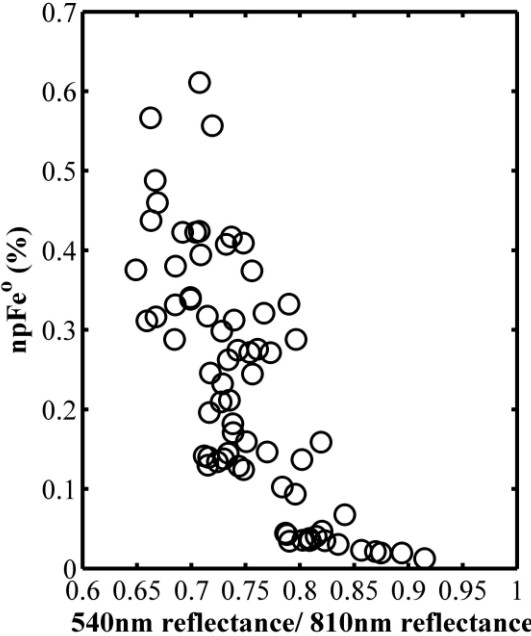

**Figure 6.** Correlation between the ratio of 540 nm reflectance/810 nm reflectance and $npFe^0$ for all size groups of LSCC soil samples.

## 5. Discussion

### 5.1. Scattering Points Analysis

Although normalizing 540 nm SSA by 810 nm SSA can partially remove the grain size effect of lunar soils and improve the correlation between $npFe^0$ abundance and the 540 nm/810 nm SSA ratio, there are still many points not tightly close to the regression line (Figure 4). This might be explained by two reasons. First, the scattering could result from the high ilmenite content of lunar soils. Trang and Lucey [38] attempted to derive $npFe^0$ abundance using Hapke's RTM and found that the variation of ilmenite content of lunar soils has a strong impact on the estimation result. This is consistent with our finding that many scattering points occur for soil samples with ilmenite content higher than 5% (Figure 7). The content of $TiO_2$, as the major composition of ilmenite, will increase in the soil samples with increasing ilmenite. Enhanced $TiO_2$ could result in a blue slope (in contrast to the reddening effect of $npFe^0$) of the reflectance spectra at short wavelengths [55,56]. Our regressed relationship as a measure of the slope for the reflectance continuum at short wavelengths could be affected by the enhanced $TiO_2$ content in lunar soils. We analyzed the correlation between $npFe^0$ and the 540 nm/810 nm SSA ratio for lunar samples of varying $TiO_2$ content. The result shows that some scattering points are samples with $TiO_2$ content higher than 6%. The correlation between $npFe^0$ abundance and the SSA ratio for lunar soils samples with $TiO_2$ content < 6% is remarkably better than ($R^2 = 0.92$) (Figure 8a) that for samples with $TiO_2$ content > 6% ($R^2 = 0.11$) (Figure 8b), indicating higher $TiO_2$ content within

lunar soils will to some extent interfere with the abundance prediction of npFe$^0$. Further correlation analysis on LSCC data suggests an obvious linear relationship between the ilmenite and TiO$_2$ content of lunar soils. 6% TiO$_2$ roughly corresponds to 5% ilmenite in lunar soils (Figure 9), which agrees well with our observation.

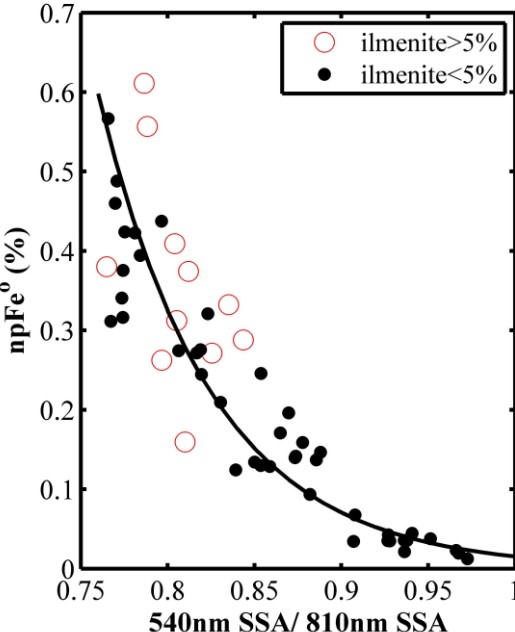

**Figure 7.** Correlation between the ratio of 540 nm SSA/810 nm SSA and npFe$^0$ denoted by different groups of LSCC lunar samples with varying ilmenite content. Solid black line is the regression line defined in Figure 4.

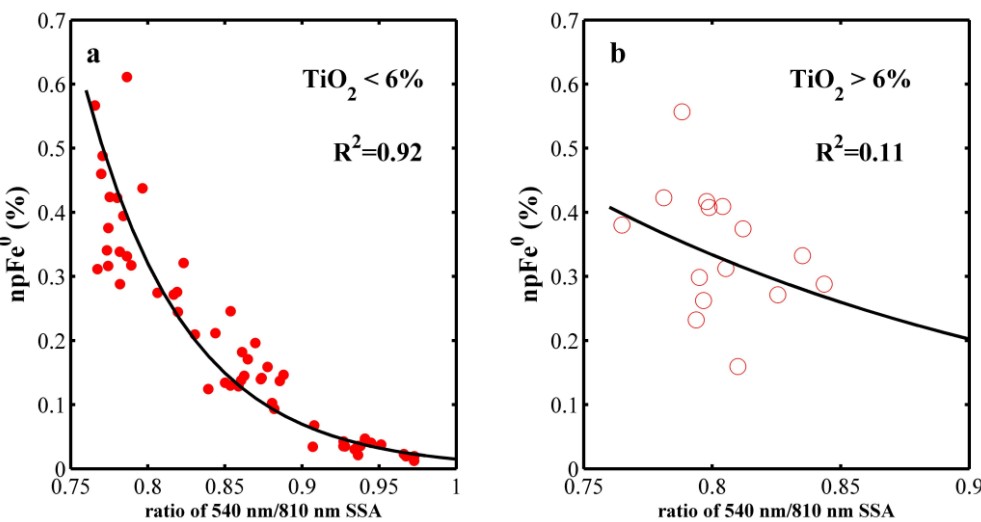

**Figure 8.** Correlation between the ratio of 540 nm SSA/810 nm SSA and npFe$^0$ for (**a**) lunar soil samples with TiO$_2$ content < 6% and for (**b**) lunar soil samples with TiO$_2$ content > 6%.

In addition to the interference from the ilmenite/TiO$_2$ content of lunar soils, some scattering points may result from samples possessing abundant (>55%) agglutinitic glass (Figure 10). Higher agglutinitic glass content in lunar soils could even cause the saturation of a regressed relationship in which the ratio of 540 nm/810 nm SSA changes very slightly with apparent increase in npFe$^0$ abundance (red circle in Figure 10). This is evidenced by the relatively higher correlation between the npFe$^0$ and SSA ratio for soil samples with agglutinitic glass < 55% (R$^2$ = 0.91) than that for soil samples with agglutinitic glass > 55% (R$^2$ = 0.78) (Figure 11). We attributed this result to the fact that in addition to the fine size npFe$^0$

coating on the surface of lunar soil grains, there are also large size metallic irons (>50 nm), termed as microphase iron or Britt-Pieters particle, residing in the interior of agglutinitic glass [38,57,58]. These larger size microphase irons mainly darken the lunar soils without much reddening effect compared to that of smaller size npFe$^0$ [58,59]. Higher abundance of large size microphase iron could accumulate in lunar soils as the agglutinitic glass content increases, resulting in a much flatter reflectance spectra of lunar soils at short wavelengths [46,49]. Therefore, the 510 nm/810 nm SSA ratio as a slope measure (reddening) at short wavelengths becomes insensitive to the variation of npFe$^0$ abundance.

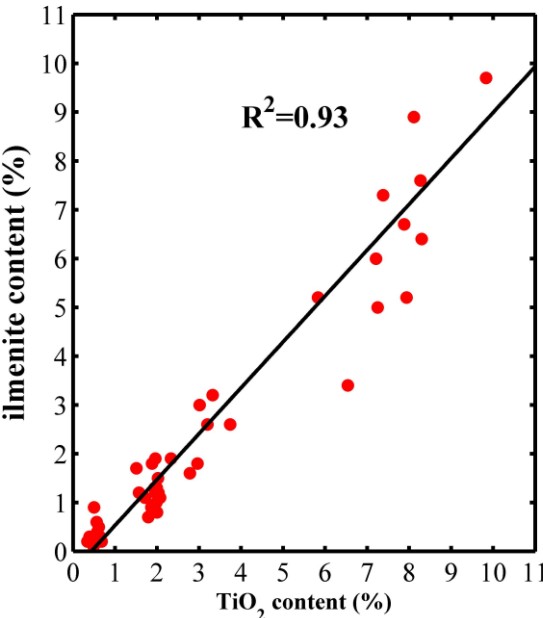

**Figure 9.** Linear correlation between ilmenite content and TiO$_2$ content of lunar soil samples.

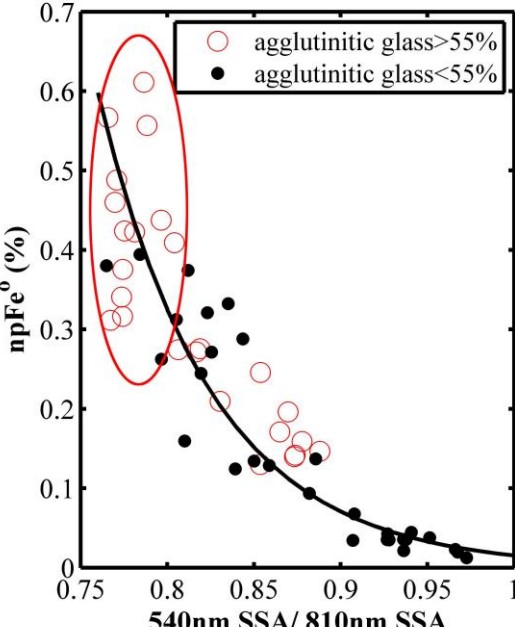

**Figure 10.** Correlation between the ratio of 540 nm SSA/810 nm SSA and npFe$^0$ denoted by different groups of LSCC lunar samples with varying agglutinitic glass content. Solid black line is the regression line defined in Figure 4.

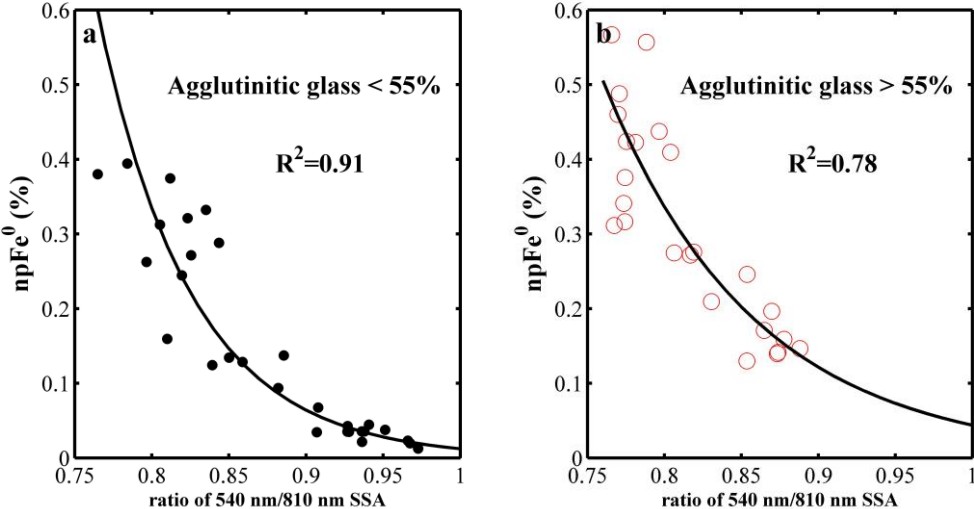

**Figure 11.** Correlation between the ratio of 540 nm SSA/810 nm SSA and npFe$^0$ for (**a**) lunar soil samples with agglutinitic glass content < 55% and for (**b**) lunar soil samples with agglutinitic glass content > 55%.

### 5.2. Non-linear Correlation Analysis

Figure 4 shows that the correlation between npFe$^0$ and the ratio of 540 nm SSA /810 nm SSA can be regressed using an exponential function rather than a linear function. One reason might be that higher agglutinitic glass content within lunar soils, as discussed in Section 5.1, could result in the unsensitivity of the 540 nm/ 810 nm SSA ratio to the variation of npFe$^0$ abundance. This can be demonstrated by the fact that a relatively acceptable linear relationship ($R^2 = 0.82$) exists between soil samples of lower agglutinitic glass content and npFe$^0$ abundance when soil samples of higher agglutinitic glass content were ignored (Figure 12a). Alternatively, it was found that if sub-mature soil samples were not considered, the correlation between npFe$^0$ and the ratio of 540 nm SSA/810 nm SSA can be also fitted by two separate linear functions corresponding to mature and fresh samples, respectively (Figure 12b). The exponential form seems to be the combination of these two linear forms. Mature soil samples show a steeper regressed slope and a lower $R^2$ compared to that of fresh soil samples. This is because the maturity of lunar soils roughly positively correlates with the their agglutinatuc glass content (Figure 13). Mature lunar soils might have an increased agglutinitic glass content relative to that of fresh soils. As discussed in 5.1, this could lead to the 540 nm/810 nm SSA changing very slightly with apparent variation in npFe$^0$ abundance. The two-segment linear fitting also implies that the maturity of lunar soils should be taken into account (e.g., optical maturity parameter: OMAT) in future work to further refine the model, and a linear correlation between npFe$^0$ abundance and the refined spectral parameter could be expected.

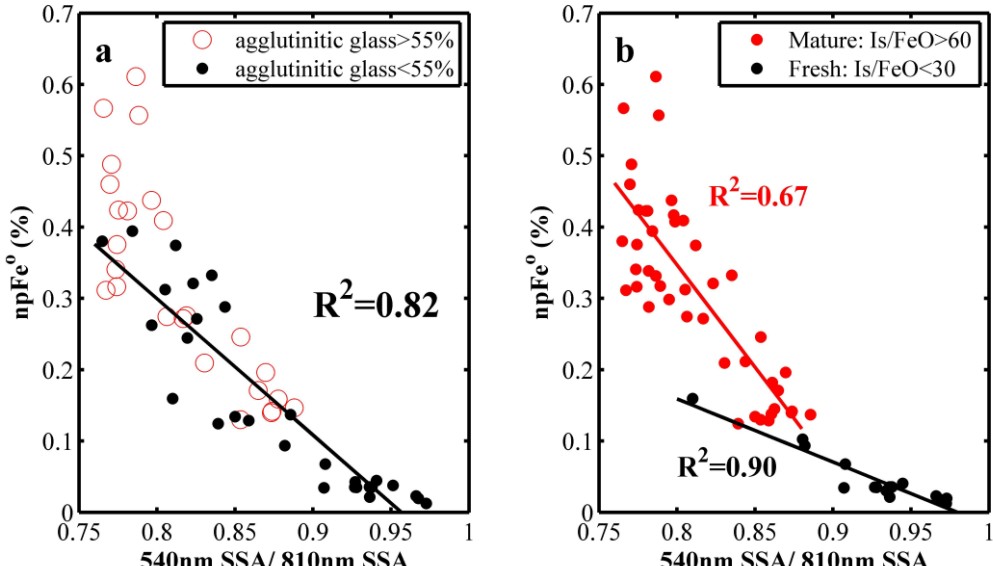

**Figure 12.** (**a**) Correlation between the ratio of 540 nm SSA/810 nm SSA and npFe$^0$ for lunar soil samples with agglutinitic glass content < 55%. Red circles are soil samples ignored in the regression process with agglutinitic glass content > 55%; (**b**) Correlation between the ratio of 540 nm SSA/810 nm SSA and npFe$^0$ for mature (red dots and line) and fresh (black dots and line) lunar soil samples.

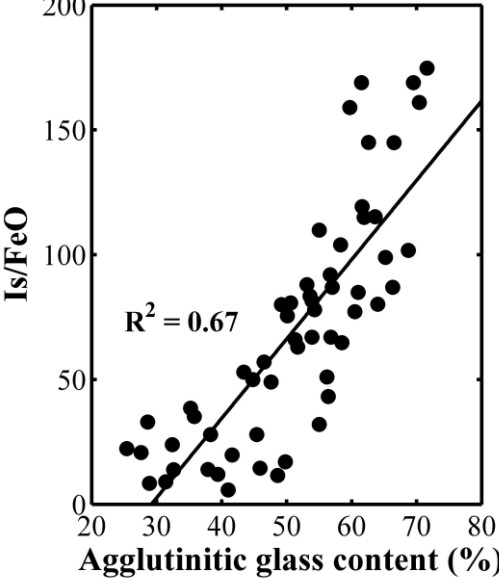

**Figure 13.** Correlation between the maturity (I$_s$/FeO) and agglutinitic glass content of LSCC lunar soil samples. Solid black line is the linear regression line.

## 5.3. Case Application and Comparison

Although accurate estimation of npFe$^0$ abundance in lunar soils based on the 540 nm /810 nm SSA ratio could be impeded by the agglutinitic glass and ilmenite content, this ratio can still be used to investigate the variation trend of npFe$^0$ for lunar surface areas. In this study, this empirical model was applied to a swirl and a mare region on the lunar surface to evaluate their regional npFe$^0$ distribution. The estimation results for the two regions were also compared, aiming at testing whether this model is sensitive to the variation of npFe$^0$ abundance in lunar soils.

1)    Lunar swirls

Lunar swirls are bright curvilinear markings on the lunar surface. They are optically immature compared to the surrounding areas and associated with regions having high magnetic field strength [60,61]. One prevailing hypothesis for the origin of lunar swirls states that the magnetic field above swirl surfaces deflects the solar wind ions away from swirl regions and directs them to the off-swirl regions. Less $npFe^0$ is generated in swirl regions than surrounding off-swirl regions owing to reduced solar wind flux [62,63]. This provides the best target area for our model evaluation.

We applied the model (exponential function in Figure 4) to one sub-image of $M^3$ level 2 reflected spectral data covering the central region of a famous lunar swirl 'Reiner Gamma' (Figure 14a) to assess whether the mapped $npFe^0$ distribution geographically agrees well with the bright swirl observed in the remote sensing image.

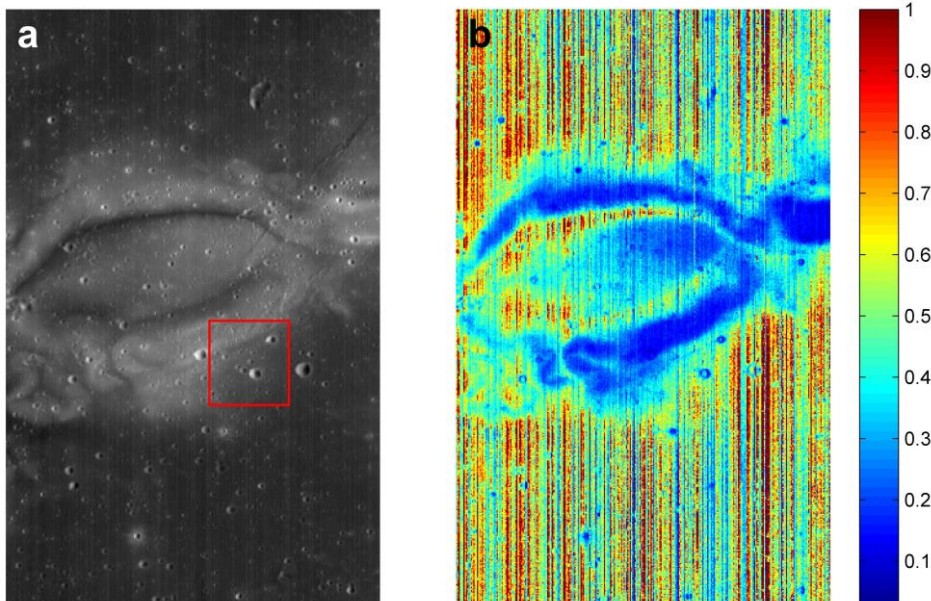

**Figure 14.** (**a**) $M^3$ 750 nm reflectance image for lunar swirl 'Reiner Gamma'; (**b**) $npFe^0$ abundance distribution of 'Reiner Gamma'.

Our results show that the swirl region evidently possesses a lower amount of $npFe^0$ relative to the off-swirl region (Figure 14b). The bright swirl in the image (Figure 14a) geographically correlates well with the lunar surface of decreased $npFe^0$, and the outline of swirl can be fully characterized by the areas deficient in $npFe^0$ abundance.

The geographical agreement between the 750 nm reflectance image and $npFe^0$ distribution map is reasonable and not an artifact resulting from higher correlation between reflectance data and their corresponding SSA. Although reflectance at 750 nm correlates with the single band SSA at 540 nm or 810 nm to some degree, it is not necessarily correlated with the ratio of 540 nm SSA/810 nm SSA. Shown in Figure 15a,b are the reflectance image at 750 nm of one sub-area in Figure 14a (red square) and the ratio image of 540 nm SSA/810 nm SSA of that sub-area, respectively. It can be seen that although some areas are positively correlated (bright area indicated by red arrows), other surface areas are negatively correlated. Parts of crater walls or floors (yellow arrow) with higher reflectance at 750 nm show a decreased ratio of 540 nm SSA/810 nm SSA, contrasting to the trend seen in the bright areas (red arrow). Some small craters in the 750 nm reflectance image (Figure 15a) even disappear in the SSA ratio image (Figure 15b) (green arrow). These observations suggest that the geographical correlation between derived $npFe^0$ distribution and 750 nm reflectance is not an artifact of the way the data reduced. Single band reflectance (e.g., 750 nm reflectance) reflects the brightness variation of the lunar surface, and it can be influenced by many factors such as variation in grain size and composing minerals of lunar soils, shadow effects, topography variation and viewing geometry. Using one band

reflectance or SSA alone cannot provide a reliable prediction of npFe$^0$ abundance (Figure 2). However, the ratio of 540 nm SSA/810 nm SSA can partially remove these effects, amplifying the npFe$^0$ effect on regulating the short wavelength spectra (Figure 4). It is capable of reflecting the regional npFe$^0$ variation trend when there is indeed npFe$^0$ abundance difference within lunar soils. The effectiveness of this ratio in npFe$^0$ estimation can also be validated by the fact that the derived abundance of npFe$^0$ within this region ranges from 0% to 1% close to the reported values of lunar soil samples [41] and the predicted values from remote sensing data [38].

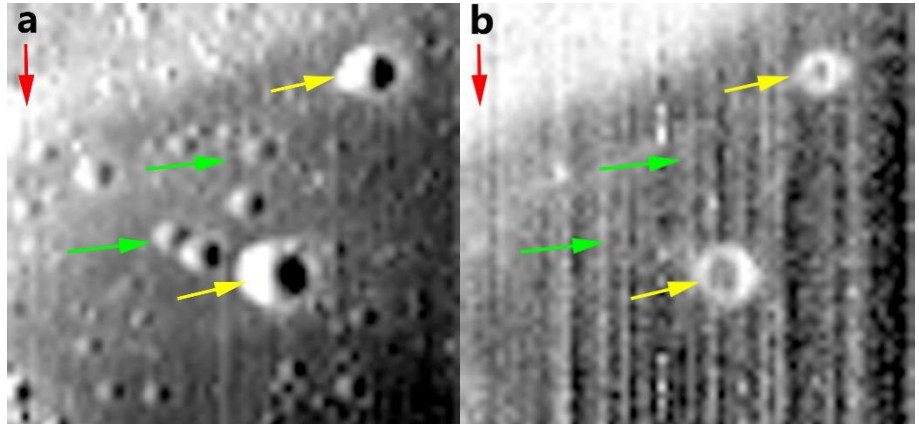

**Figure 15.** (**a**) M$^3$ 750 nm reflectance image of one sub-area in Figure 14a (red box); (**b**) the ratio of 540 nm/810 nm SSA of this sub-area. Red arrow shows positively correlated bright swirl regions between the two images. Yellow arrows show some parts of crater wall or floor having higher brightness in the 750 nm reflectance image while having low values in the 540 nm/810 nm SSA ratio image. Green arrows show the craters disappeared in the 540 nm/810 nm SSA ratio image.

2)    Lunar maria

To make a comparison with the swirl region, distribution of npFe$^0$ of a mare region to the north of 'Reiner Gamma' was also investigated. This mare region is relative homogenous without apparent albedo difference across its surface, except some distributed fresh impact craters (Figure 16a). It is expected that the npFe$^0$ variation for this region should not be as strong as that of the lunar swirl (Figure 14b).

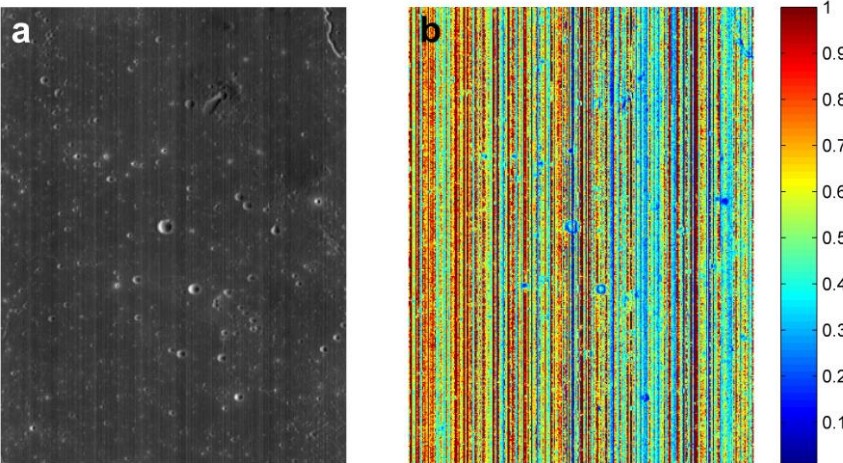

**Figure 16.** (**a**) M$^3$ 750 nm reflectance image of the mare region to the north of Reiner Gamma; (**b**) npFe$^0$ abundance distribution.

By applying this empirical model to this mare region, it was found that, in contrast to the large npFe[0] abundance difference between the swirl and off-swirl regions in Figure 14b, no surface area with obvious decreased or increased npFe[0] can be observed within this mare region (Figure 16b). The npFe[0] distribution of the whole region only bears resemblance to that of the off-swirl regions in Figure 14b.

It is worth noting that even in such areas without significant npFe[0] variation, the fresh craters on the surface of the mare can still be clearly revealed by regional distribution of npFe[0] (blue round spots or rings in Figure 16b). These fresh craters show a decreased amount of npFe[0] consistent with what is expected on the lunar surface. Because of the short exposure time to a space weathering environment, a lower amount of npFe[0] could have accumulated in these craters relative to the surrounding lunar soils [43]. This also demonstrates the effectiveness of this empirical model in estimating npFe[0] abundance of the lunar surface. However, many strips can be seen in both investigated swirl and mare regions (Figures 14b and 16b). This can be ascribed to the noise originating in the original M[3] data (strips in Figures 14a and 16a) rather than a true variation of npFe[0] abundance.

## 6. Conclusions

The 540 nm/810 nm SSA ratio is highly correlated to npFe[0] abundance in lunar soils. This spectral ratio highlights the spectral importance of npFe[0], reduces the grain size effect of lunar soils on the short wavelengths reflectance spectra and, thus, can be considered as a useful spectral index for the prediction of npFe[0] abundance within lunar soils.

However, the performance of this simple empirical model does show some interference from the high content of agglutinitic glass and ilmenite in lunar soils. More work regarding removing the effects of high agglutitinic glass and ilmenite/$TiO_2$ content should be considered in future work to further refine the model's performance in estimating npFe[0] abundance of lunar soils.

**Author Contributions:** Data curation, B.L. and R.X.; Formal analysis, D.L.; Investigation, G.Z.; Methodology, D.L.; Project administration, C.L.; Resources, B.L.; Validation, X.R. and C.L.; Writing—original draft preparation, D.L.; writing—review and editing, Y.Z. All authors have read and agreed to the published version of the manuscript.

**Funding:** This work was supported by the National Natural Science Foundation of China (41601374, 11941002 and 61605231) and by the B-type Strategic Priority Program of the Chinese Academy of Sciences (XDB41000000).

**Acknowledgments:** We thank Kevin Cannon and an anonymous reviewer for their constructive reviews and comments, which greatly improved this manuscript. We also thank the Reflectance Experiment Laboratory (Relab) of Brown University and Planetary Geoscience Institute of the University of Tennessee for making the LSCC dataset publicly available.

**Conflicts of Interest:** The authors declare no conflicts of interest.

## Appendix A

An approximate solution to the radiative transfer equations was proposed by [10,54,64] to describe the light scattering behavior of lunar soils based on the assumption that lunar surface materials are intimately mixed and lunar soil grains are closely spaced with their size larger than spectral wavelengths. In his work, the reflectance spectra of lunar soils can be converted to its SSA by the following equation:

$$\mathrm{R} = \frac{\omega}{4}\frac{1}{\mu_0 + \mu}\{[1 + \mathrm{B}(g)] \times \mathrm{P}(g) + \mathrm{H}(\mu_0, \omega) \times \mathrm{H}(\mu, \omega) - 1\} \tag{A1}$$

Here, R is remotely obtained or lab-measured reflectance, $\omega$ is the SSA of lunar soils, $\mu_0$ and $\mu$ are the cosine of the incidence light angle (i) and emittance light angle (e), respectively. g is the phase angle. B(g) and P(g) are the backscattering function and phase function of lunar soils, and H represents the multiple scattering effect between lunar soil grains.

It was found that the backscattering function B(g) defining the opposition effect of lunar soils decreases significantly with increasing phase angle g and can be ignored when g is greater than 15° [54]. Because all the reflectance spectra of the soil samples used in this study were measured at phase angle

30° and most of the remotely obtained reflectance spectra such as $M^3$ data had also been photometrically corrected to the phase angle 30°, B(g) was set to be zero to ignore the backscattering effect of lunar soils. To further simplify the model, the lunar soils were considered as isotropic scattering and the phase function P(g) was set to be unity in this study. This assumption was adopted and supported by the work [65], which found that it will not strongly affect the performance of Hapke's RTM. According to these assumptions, Equation (A1) can be simplified as follows:

$$R = (\omega \times H(\mu_0, \omega) \times H(\mu, \omega)) / (4 \times (\mu + \mu_0)) \tag{A2}$$

Here, $H(x, \omega)$ is Chanderasekhar's function for isotropic scattering of lunar soils, and x represents $\mu_0$ (incident light angle) or $\mu$ (emittance light angle) [50].

Setting the incidence angle and emission angle as 30° and 0°, the same as the measured reflectance data of lunar soil samples in this study, the SSA of lunar soils could be derived from Equation (A1) and Equation (A2) as follows [50]:

$$\omega = 1 - \left( \frac{-27.865R + \sqrt{4.029R^2 + 602.932R + 268.696}}{51.71R + 16.392} \right)^2 \tag{A3}$$

These steps are the same as the work of Yan et al. [50].

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
