# Peer review of "An Empirical Model to Estimate Abundance of Nanophase Metallic Iron (npFe0) in Lunar Soils"

_remotesensing, doi:10.3390/rs12061047_

Round 1

Reviewer 1 Report

This is an interesting idea, and I do not see any major flaws in reasoning. However, there are some details that I think should be included, some assumptions that should at least be acknowledged, and this does require fairly extensive editing for English style.

Detailed comments:

Exactly where did the spectroscopy data come from? Two papers that have spectra in them are cited (citations #7 and #15), but those papers don’t include digital versions of the spectra, at least as far as I could tell. Were the spectra just digitized from the journal figures, or (I hope) was the original data acquired from somewhere? A more explicit reference to the source of the data should be given.

The authors have calculated the npFe0 by using equation 1. This includes the Is/FeO value. However, the values given by the LSSC are for the bulk <250 µm fraction, so if I understand it right, the authors assume that Is/FeO is the same for all grain sizes. Is there any data in the literature to support this? If not, it may be a good assumption, but this should be pointed out as an assumption.

Line 214: How difficult is it to convert reflectance to SSA? Is it simple to do, or are there assumptions that need to be made that would be more difficult for spacecraft data than for the laboratory data used here. I really don’t know the answer (I’m not a spectroscopist), but I think it needs to be addressed, at least briefly.

In 4.3, I would suggest doing a fit and determining an R2 for the high-Ti and low-Ti samples, and for the samples with high and low agglutinate abundances. Particularly for the Ti, the Trang and Lucey work shows that while it’s difficult, it might be possible to derive Ti content from spectra, so if that could be quantified, this technique would be more valuable.

In the analysis of swirls (4.4,1) what the authors have really shown is that the reflectance at 750 nm correlates with the ratio of SSA at 540 nm to that at 810 nm (since that is proportional to the npFe0 in equation 1). Assuming that reflectance generally (though not perfectly) correlates with SSA, is this a surprise? I’m not convinced this is a good test. At the least, this discussion should include a discussion to convince the reader that the correlation is not just an artifact of the way the data is reduced. To me, 4.4,2 is more convincing, showing that the mare region has little variation except for fresh craters.

Some minor comments:

I think that the introduction of any paper referring to nanophase iron should at least cite the recent experimental results by Michelle Thompson, even though those results are sometimes counterintuitive. The best references are Thompson et al. (2016) Meteoritics & Planetary Science 51, 743-756; and Thompson et al. (2017) Meteoritics & Planetary Science 52, 413-427

It is better to call it “nanophase metallic iron” than “native nanophase metallic iron”, because “native” iron means that it is metallic.

In reference 4, the first author’s name is “Noble”, not “Nobel”

Reviewer 2 Report

Summary

This article presents a new empirical model to estimate the abundance of nanophase iron in lunar soils. Using single-scattering albedo values from the Lunar Soil Characterization Consortium, it proposes a relation between the npFe0abundance and the ratio of SSA at 540 to 810 nm. This new calibration is then applied to two test cases using M3data: one of a lunar swirl, and another from a mare region.

Overall, the article is straightforward and well-written. I support publication in Remote Sensing after the following comments are addressed.

If the authors have any questions, they are welcome to contact me: Kevin Cannon (cannon@ucf.edu).

General Comments

  1. Although the article points out deficiencies in previous models for npFe abundance (lines 77-92), I do not feel it does enough to put the new empirical model into the context of previous work. Specifically, is the empirical model expected to be more accurate than the npFe0maps from Trang and Lucey (2019)? Or is the main advantage that longer wavelengths are not required?
  2. The <45 micron fraction of the LSCC samples is used as an analog for bulk soil. Is the <45 micron fraction the coarsest fraction that was examined in the LSCC? 45 microns is about the mean grain size of lunar soil, which means that half of the grains will be larger than this. How confident are you that the <45 micron model parameters will apply to unsieved regolith on the Moon?

Line-by-line Comments

Line 39: “Moon” should be capitalized. Change throughout the manuscript.

Line 40: The airless body Itokawa has returned samples, the Moon is not the only body.

Line 44: Change “and an attenuated absorption band” to “and attenuated absorption bands”

Line 48: Change “a long time exposure of fresh lunar soils to space weathering environment” to “a long duration exposure of fresh lunar soils to the space weathering environment”

Line 52: Change “assessing mineralogy of lunar surface” to “assess the mineralogy of the lunar surface”

Line 53: Change “space weathering effect” to “space weathering effects”

Line 62: Change “establishing” to “establish”

Line 78: Change “not sufficient” to “insufficient”

Line 89: Change “mineral absorption” to “mineral absorptions”

Line 96: Change “size effect” to “size effects”

Line 97: Change “estimation is” to “estimation are”

Table 1: I suggest adding two additional columns to this table: 1 column for the grain size, instead of using superscript letters (a,b,c,d) as you do now. And another column to indicate if the soils are mare or highlands.

Line 111: Change “of lunar soil” to “of lunar soils”

Line 117: Change “convergent” to “converge”

Line 126: Change “developed” to “develop”

Line 172: Is there a physical basis to expect an exponential form for the equation? For example, does anything in Hapke theory or Mie theory suggest you would end up with such a relationship? It would be useful to add a sentence or two to address this.

Line 266: Change “marking” to “markings”

Line 266: Change “optically immature than” to “optically immature compared to”

Line 270: Change “A less amount of npFe0are” to “Less npFe0is”

Line 278: Change “Bright swirl in” to “The bright swirl in”

Line 280: Change “is capable in” to “is capable of”

Line 302: You might be able to reduce the noise and striping by using a “kernel”, where instead of a single wavelength band for 540 and 810 nm, you take an average of that band and the surrounding bands beside it. For M3you could use an average of 540 and 580 nm, and an average of 790, 810 and 830 nm. It is worth trying to see if it reduces the noise. See Viviano-Beck et al. 2014 (https://agupubs.onlinelibrary.wiley.com/doi/pdf/10.1002/2014JE004627) for more details on this.
